# The Impact of Cold Ischaemia Time on Outcomes of Living Donor Kidney Transplantation: A Systematic Review and Meta-Analysis

**DOI:** 10.3390/jcm11061620

**Published:** 2022-03-15

**Authors:** Stijn C. van de Laar, Jeffrey A. Lafranca, Robert C. Minnee, Vassilios Papalois, Frank J. M. F. Dor

**Affiliations:** 1Imperial College Renal and Transplant Centre, Hammersmith Hospital, Imperial College Healthcare NHS Trust, London W12 0HS, UK; s.c.vandelaar@erasmusmc.nl (S.C.v.d.L.); j.a.lafranca@gmail.com (J.A.L.); vassilios.papalois@nhs.net (V.P.); 2Division of HPB and Transplant Surgery, Department of Surgery, Erasmus MC Transplant Institute, 3015 CN Rotterdam, The Netherlands; r.minnee@erasmusmc.nl; 3Department of Surgery and Cancer, Imperial College London, London SW7 2AZ, UK

**Keywords:** kidney transplantation, cold ischemia time, living donors

## Abstract

Studies have been carried out to investigate the effect of a prolonged cold ischaemia time (CIT) on the outcomes of living donor kidney transplantation (LDKT). There is no clear consensus in the literature about the effects of CIT on LDKT outcomes, and therefore, we performed a systematic review and meta-analysis to provide evidence on this subject. Searches were performed in five databases up to 12 July 2021. Articles comparing different CIT in LDKT describing delayed graft function (DGF), graft and patient survival, and acute rejection were considered for inclusion. This study is registered with PROSPERO, CRD42019131438. In total, 1452 articles were found, of which eight were finally eligible, including a total of 164,179 patients. Meta-analyses showed significantly lower incidence of DGF (odds ratio (OR) = 0.61, *p* < 0.01), and significantly higher 1-year graft survival (OR = 0.72, *p* < 0.001) and 5-year graft survival (OR = 0.88, *p* = 0.04), for CIT of less than 4 h. Our results underline the need to keep CIT as short as possible in LDKT (ideally < 4 h), as a shorter CIT in LDKT is associated with a statistically significant lower incidence of DGF and higher graft survival compared to a prolonged CIT. However, clinical impact seems limited, and therefore, in LDKT programmes in which the CIT might be prolonged, such as kidney exchange programmes, the benefits outweigh the risks. To minimize these risks, it is worth considering including CIT in kidney allocation algorithms and in general take precautions to protect high risk donor/recipient combinations.

## 1. Introduction

The best therapy for patients with end-stage renal disease (ESRD) is kidney transplantation. Not only does kidney transplantation reduce the risk of morbidity and mortality and is associated with a better quality of life, but it also has superior outcomes compared to other forms of renal replacement therapy [1,2,3]. Unfortunately, kidney transplantation is not readily available for all ESRD patients in need due to donor organ shortage; although the waiting list for patients with ESRD is decreasing and the number of transplantations is increasing, only around 46% of patients on the waiting list received a donor kidney in 2019–2020 in the United Kingdom. Furthermore, during this period, 285 ESRD patients (4%) did not survive whilst on the waiting list [4]. To counter this problem, several therapies and strategies have been implemented to improve the availability of kidney grafts and the number of transplantations.

Over the years, living donor kidney transplantation (LDKT) has proved superior to deceased donor kidney transplantation (DDKT) [5,6,7], especially as it facilitates pre-emptive transplantation [8]. One of the other many benefits of LDKT is that the cold ischaemia time (CIT) is shorter since generally donor and recipient operations happen in the same centre and on the same day. Transplant professionals feel that this is important since a shorter CIT leads to what is believed to be a better outcome in the recipient [9,10,11]. This principle is based on the outcomes of CIT in the deceased donor kidney transplantation literature; we know that a short CIT is associated with better transplant outcomes [9,10], and prolonged CIT is strongly correlated with a higher incidence of delayed graft function (DGF) [12,13,14,15] and a possible impact of CIT leading to a higher incidence of graft failure and graft loss [12,16,17,18,19].

To expand the options of having LDKT for patients with incompatible donors, kidney exchange programmes were introduced. The first national Kidney Exchange Programme (KEP) in Europe was the Crossover Kidney Transplantation Programme started in 2003 in the Netherlands [20,21]. It uses an algorithm to provide live donor kidneys to people with ESRD that cannot receive a kidney from their potential donor because of AB0 incompatibility or donor-specific antibodies. In April 2007, the United Kingdom initiated the UK Living Kidney Sharing Scheme (UKLKSS) to match incompatible recipients to an appropriate kidney donor [22]. In this system, the donor kidney might be transported from one centre to another, which is associated with longer anticipated CIT, in contrast to kidney exchange programmes in the Netherlands and Canada, where the donor travels to the recipient centre where both live donor nephrectomy and transplant operations take place with CIT comparable to direct LDKT.

To investigate and understand the impact of a longer CIT in LDKT, a comprehensive review is needed to analyse the available data since literature on this topic is limited. With the current study, we hope to provide the highest level of evidence and recommendations.

## 2. Materials and Methods

All aspects of the Cochrane Handbook for Interventional Systematic Reviews were followed, and the manuscript was written according to the PRISMA statement [23,24]. The systematic review protocol has been registered with the PROSPERO International prospective register of systematic reviews database (PROSPERO registration number: CRD42019131438, https://www.crd.york.ac.uk/PROSPEROFILES/131438_PROTOCOL_20190407.pdf, accessed on 12 July 2021)

### 2.1. Literature Search Strategy

Comprehensive searches were carried out in Embase, MEDLINE OvidSP, CENTRAL (The Cochrane Library 2019, Issue 2), Web of Science and Google Scholar 100 top-ranked comparing prolonged CIT to short CIT on outcomes of LDKT. The search was performed for articles published up to 12 July 2021. Search terms for each search engine are provided in Appendix B. Manual reference checks in included papers were performed to check for potentially missing studies. 

### 2.2. Outcome Parameters

Our initial outcome parameters were clinical parameters consisting of the following: DGF (defined as the need for dialysis in the first week after transplantation), 1- and 5-year graft survival, 1- and 5-year patient survival, acute rejection, serum creatinine, urine output, and graft function (expressed as estimated glomerular filtration rate (eGFR)). 

### 2.3. Literature Screening

Eligibility was independently assessed for each study by two reviewers (SCvdL, JAL). First, studies were in- or excluded based on title and abstract. If a paper was included, the full text was read and considered whether the study met the inclusion criteria. We used predefined exclusion criteria: case reports, letters, editorials, case series, animal studies, paediatric studies (under the age of 18), papers not written in English, or if the abstract revealed no relevance to the subject. Second, studies analysing one or more of the outcome measures in different CIT categories were included. By this, we mean that studies should compare CIT at different time periods, for instance: 0–2 h, 2–4 h, 4–6 h, etc., or use of a predefined cut-off point of CIT. If any discrepancies in inclusion or exclusion occurred, a senior investigator (FJMFD) was consulted.

### 2.4. Data Extraction and Critical Appraisal

Risk of bias was assessed with the risk of bias in non-randomized studies of interventions (ROBINS-I) tool [25]. The level of evidence of each outcome was established using the GRADE tool [26]. The GRADE approach defines the quality of a body of evidence by considering the within-study risk of bias (methodological quality), the directness of evidence, heterogeneity, the precision of effect estimates, and the risk of publication bias (Appendix C). 

### 2.5. Statistical Analysis

Meta-analyses were performed in line with recommendations from the Cochrane Collaboration and Meta-analysis of Observable Studies in Epidemiological guidelines [27] and were performed using Review Manager version 5.3 (The Nordic Cochrane Centre, Copenhagen, Denmark) [28]. Continuous variables in baseline characteristics were reported with the group mean weighted for the number of patients. Significance was calculated via unpaired *T* test with normal distribution assumed based on the range or standard deviation. We used random-effects models to account for possible clinical heterogeneity. Overall effects were determined using the Z test; 95% confidence intervals were given, and *p* < 0.05 was considered statistically significant. The Mantel–Haenszel analysis method was used with calculation of the overall effect using the Z test. Three methods assessed heterogeneity between studies. First, a Tau2 test and a χ^2^ test were performed for statistical heterogeneity, with a *p* < 0.1 being considered statistically significant. Second, I^2^ statistics were used to assess clinical heterogeneity, where an I^2^ of 0% to 40% is considered low heterogeneity, 30% to 60% as moderate heterogeneity, 50% to 90% as substantial heterogeneity, and 75% to 100% as considerable heterogeneity [24].

## 3. Results

### 3.1. Literature Search Results

In the initial search, we identified 1452 articles, and after deduplication, 931 unique articles remained. Subsequently, after screening titles and abstracts, 91 articles remained for full text screening. We identified eight comparative studies for our systematic review across three continents (Europe, North America, and Australia) with a total of 164,179 patients [11,29,30,31,32,33,34,35]. Five studies were included in the quantitative synthesis [11,29,30,31,35] and three studies were excluded because they did not provide a range in CIT [32,33] or used a continuous CIT instead of CIT intervals [34]. Figure 1 represents the PRISMA flow diagram for systematic reviews, and study characteristics included in this systematic review are described in Table 1. The baseline characteristics of the patients included in the meta-analysis are shown in Table 2.

### 3.2. Outcomes 

#### Delayed Graft Function

Gill et al. [11], Krishnan et al. [30], Nassiri et al. [29], Nath et al. [31], and Simpkins et al. [35] all analysed the incidence of DGF in different CIT intervals. These five studies combined provided a patient population of 94,693 (92.65%) patients with a CIT of less than four hours and 7507 (7.35%) with more than four hours. All studies investigated delayed graft function, and all except one found a significant lower odds ratio (OR) for a CIT of less than four hours (OR = 0.61, 95% CI, 0.49 to 0.77, *p* < 0.0001) (Figure 2). In the short CIT group, 4293 (4.53%) patients presented with DGF, while the prolonged CIT cohort included 508 (6.77%) patients with DGF. 

In different analyses, we compared different CIT time-groups to a reference CIT of zero to two hours. This led to the following OR: compared to two to four hours: OR = 0.89 (95% CI, 0.82 to 0.97, *p* = 0.01), compared to four to six hours: OR = 0.54 (95% CI, 0.47 to 0.62, *p* < 0.001) and compared to six to eight hours: OR = 0.48 (95% CI, 0.41 to 0.56, *p* < 0.001) (Figure 3). 

Two studies could not be included in the meta-analysis: the study by Redfield et al. [32] showed that transplants of patients who had DGF had a longer CIT (2.6 versus 2.2 h, *p* < 0.001) compared to patients who did not develop DGF. Nath et al. [31] found that patients in the four to eight hours CIT group had a significantly higher incidence of DGF than those with a CIT of less than two hours, with a mean difference of 3.1 (95% CI 0.8 to 5.5) per cent (*p* = 0.007). Figure 4 shows a graphical representation of the relation between CIT and incidence of DGF. 

### 3.3. Graft Survival

#### 3.3.1. One Year

Nath et al. [31] and Simpkins et al. [35] studied 1-year graft survival in different CIT intervals. These two combined resulted in 44,696 (93.45%) patients with a CIT of less than four hours and 2927 (6.55%) patients with a CIT of more than four hours. A CIT of less than four hours is in favour in terms of 1-year graft survival with an OR = 0.72 (95% CI, 0.60 to 0.87, *p* < 0.001) (Figure 5). A reference CIT of 0–2 h compared to 2–4 h of CIT showed no significant difference: OR = 0.89 (95% CI, 0.78 to 1.02, *p* = 0.09. Further, a prolonged CIT of 4–8 h compared to the reference led to an OR = 0.84 (95% CI, 0.72 to 0.97, *p* = 0.02) (Figure 6).

#### 3.3.2. Five Years

Krishnan et al. [30], Nath et al. [31], Nassiri et al. [29], and Simpkins et al. [35] investigated 5-year graft survival in different CIT intervals. These four studies included 48.757 (94.28%) patients with a CIT of less than four hours and 4.945 (6.14%) patients with more than four hours of CIT.

All four studies favoured a CIT shorter than four hours compared to a CIT longer than four hours: OR = 0.88 (95% CI, 0.79 to 0.99, *p* = 0.04) (Figure 7).

In different analyses, we compared different CIT time-groups to a reference CIT of zero to two hours. This led to the following ORs: compared to a CIT of 2–4 h: OR = 1.04 (95% CI, 0.86 to 1.27, *p* = 0.82), and compared to a CIT of 4–8 h: OR = 0.88 (95% CI, 0.77 to 1.00, *p* = 0.04) (Figure 8).

### 3.4. Five-Year Graft Survival Hazard Ratio

Some studies did not include absolute numbers for 5-year graft survival in their results, but provided hazard ratios only, i.e., Nath et al. [31], and therefore we provided an extra forest plot with these results. Krishnan et al. [30], Nath et al. [31], Simpkins et al. [35], and Gill et al. [11] all investigated the hazard ratio of zero to two hours of CIT (reference) versus two to four and four to eight hours of CIT. In the zero to two hours versus two to four hours of CIT, we found an HR = 1.04 (95% CI, 0.98 to 1.09, *p* = 0.22) for graft failure (Figure 9). In the zero to two hours versus four to eight hours of CIT, we found a HR = 1.20 (95% CI, 1.06 to 1.37, *p* = 0.006) for graft failure, both in favour of a shorter CIT (Figure 10).

### 3.5. Ten-Year Graft Survival

Initially, we did not include ten-year graft survival as an outcome measure. However, we found that two studies included graft survival follow-up up to ten years [30,35]. A CIT of less than four hours was not associated with a significant higher graft survival at ten-year follow-up, OR = 0.84 (95% CI, 0.64 to 1.09, *p* = 0.19) (Figure 11).

### 3.6. Patient Survival

Two studies provided patient survival data for the meta-analysis. Krishnan et al. [30] and Nath et al. [31] included 1-year patient survival. These two studies combined a total of 10,753 (83.88%) patients in the CIT shorter versus four hours group and 2066 (16.12%) patients in the CIT longer versus four hours group. We found a non-significant OR = 0.74 (95% CI, 0.51 to 1.07, *p* = 0.11) in favour of the shorter CIT (Figure 12). Two studies [30,31] analysed 5-year patient survival. These two studies combined included a total of 10,807 (83.95%) patients in the CIT shorter than four hours group and 2.66 (16.05%) patients in the CIT longer than four hours group. Similar to 1-year patient survival, there was no significant difference in patient survival between both groups (Figure 13): OR = 0.54 (95% CI, 0.20 to 1.45, *p* = 0.22). 

### 3.7. Acute Rejection

Krishnan et al. [30], Nath et al. [31], and Simpkins et al. [35] all investigated the impact of CIT on acute rejection. One article described acute rejection as ‘any rejection or multiple rejections’, while the other articles do not mention the definition of acute rejection used. These three studies combined included a total of 48.189 (93.86%) patients in the CIT shorter than four hours CIT group and 3.151 (6.14%) in the more than four hours CIT group. The risk of acute rejection was not significantly different between kidney transplantations when comparing a CIT shorter or longer than four hours: OR = 1.17 (95% CI, 0.86 to 1.58, *p* = 0.31) (Figure 14).

### 3.8. Kidney Function

Three studies [30,31,35] reported kidney function with no study eligible for inclusion in the meta-analysis: Krishnan et al. showed that CIT > four hours was associated with a significantly lower 1-year glomerular filtration rate (GFR): with a CIT of more than four hours as reference compared to a CIT of 1–2 h led to an OR = −5.79 (95% CI, −9.96 to −1.62, *p* = 0.010), and when compared to a CIT of 3–4 h, this led to an OR = −5.12 (95% CI, −9.38 to −0.87, *p* = 0.017), both in favour of the CIT shorter than four hours. Nath et al. compared average creatinine levels at 1 year and found that there was no significant difference in 12-month creatinine values between 0–2 h and 2–4 h of CIT (*p* = 0.094), but patients in the 4–8 h group had significantly higher values than those with a CIT of less than two hours, with a mean difference of 3.1% (95% CI, 0.8 to 5.5, *p* = 0.007). The adjusted average creatinine levels at 12 months were 122, 124 and 126 µmol/l in the groups with a CIT of less than 2, 2–4 and 4–8 h, respectively. Simpkins et al. found that there were no statistically significant differences in 1-year serum creatinine level between recipients in any of the groups with CIT >2 h and the reference group (*p* > 0.05 for all comparisons).

### 3.9. Impact of CIT in Kidney Exchange Programmes (KEP)

Three studies [11,29,30] analysed a subgroup of patients transplanted via a kidney exchange programme. Krishnan et al. [30] compared, in a sub-analysis, the shipped Australian paired kidney exchange (AKX) (*n* = 33) versus the non AKX (*n* = 1541). Shipping of the kidney was not associated with DGF (OR (95% CI): 1.40 (0.88–2.40)) and death censored graft survival up to eight years: HR (95% CI) 0.70 (0.46–1.08). Gill et al. [11] included data from the Scientific Registry of Transplant Recipients in the United States and compared shipped (*n* = 772) versus non-shipped (*n* = 1651) KEP transplants. In their study, they found no difference between non-shipped and shipped KEP in terms of rejection, DGF, death censored graft loss and all-cause mortality. Nassiri et al. [29] included LDKT facilitated through KEP and compared a CIT ≥ 16 h (*n* = 141) to a CIT < 16 h (*n* = 2222); no significant increased odds ratio in terms of DGF (OR (95% CI: 1.199 (0.585 2.457), *p* = 0.62) and death censored graft failure (OR (95% CI): 0.353 (0.087–1.429), *p* = 0.14) were found.

### 3.10. Quality and Risk of Bias Assessment

All studies had moderate risk of bias according to the ROBINS-I tool because it was not possible to completely rule out bias due to confounding in domain I. On almost all other domains, the studies scored low risk of bias, which means the concluding risk of bias was moderate in all studies (Appendix A). The certainty of outcome and summary of findings, measured via the GRADE-tool, ranged from low to high. Most studies were assessed as a moderate certainty of outcome and downgraded for various reasons; these are shown in Appendix C.

### 3.11. Publication Bias

We used contour-enhanced funnel plots to investigate the presence of possible publication bias. In most funnel plots, no evidence of publication bias was found regarding the outcomes (Appendix A). Only with regard to acute rejection and 5-year patient survival is there a minimal level of statistical publication bias. 

## 4. Discussion

With this meta-analysis, we aimed to assess the impact of CIT on graft function and graft survival in LDKT. The discussion about CIT and its potential influence on graft and patient outcomes of LDKT is topical in a decade where ideas about nationwide, and on a small scale even international, LDKT exchange programmes are becoming more ambitious than ever before [36,37]. In the UK, for example, LDKT from the UKLKSS account for 14% of all LDKT [38]. With kidneys from living donors travelling from donor to recipient centres across and between countries, long distances are being bridged, CIT will inevitably increase. In deceased donor kidney transplantation, prolonged CIT is proven to be associated with worse transplant outcomes, higher incidence of DGF and a higher incidence of graft failure and graft loss [9,10,12,13,14,15,16,17,18,19].

The literature thus far has always stated that CIT is not a clinically relevant issue in LDKT and that living donor kidneys can tolerate even prolonged CIT of >16 h when shipped between states in the USA for example [29]. Besides an increased importance of kidney paired exchange programmes with longer CIT compared to direct LDKT, living donor kidneys include more expanded criteria donors which may impact graft outcomes [39]. These expanded criteria donor kidneys might be more sensitive to prolonged CIT compared to standard criteria living donor kidneys. 

LDKT recipients in kidney paired exchange programmes implicitly are higher risk recipients compared to recipients in non-kidney paired exchange LDKT; the KEP group consists of more highly sensitized patients, higher donor and recipient age, more human leucocyte antigen (HLA) antibodies, more retransplantations, longer waiting time before transplantation, and therefore a longer period of dialysis and less pre-emptive transplantation [38]. This theoretically would result in a higher risk of complications, impacting graft and patient survival. However, the study found no difference in graft survival between KEP and non-KEP, only a significant difference in DGF. When focussing on CIT, we recently found (van de Laar et al. [38]) in a large UK cohort (*n* = 9967) that graft survival remains excellent even in the group with a CIT > 10 h in LDKT and found a beneficial impact of a CIT of 0–2 h in non-KEP LDKT. The incidence of DGF in this study was, comparable with our findings in the meta-analysis, significantly higher in LDKT with a prolonged CIT. 

Our data show an adverse effect of a CIT longer than four hours on the risk of DGF and a decrease in the 1- and 5-year graft survival of kidney grafts after LDKT significantly, favouring a CIT of less than four hours. To put those outcomes into perspective: the numbers needed to treat would be 50 for DGF and 35 for 5-year graft survival. We have demonstrated a marginal impact of CIT on graft survival and graft quality and hope that these findings will contribute to the expansion of KEPs and provide evidence to support international collaboration in KEPs. We should keep in mind that patients or transplant professionals should not be discouraged to accept a kidney with prolonged CIT, and results should not lead to fewer transplant opportunities for patients. As shown in the study above [38], results of LDKT outperform deceased donor kidneys even with prolonged CIT. However, these outcomes still might be optimized since a higher incidence of DGF leads to a longer hospital stay which may result in less quality of life for the patient and increased costs per kidney transplant such as in DDKT [40,41]. 

Multiple strategies could be implemented to shorten the CIT in LDKT; the most effective would be simultaneous live donor nephrectomy and LDKT in case of direct donation. In this case, there is hardly any clinically relevant CIT. However, many centres cannot organize this, given the need for two theatres which are not utilized at maximum efficiency to enable simultaneous donor and recipient surgery, and this is the reason most centres default to consecutive living donor nephrectomy and LDKT. In case of kidney paired exchange, there are other differences besides simultaneous versus consecutive surgery. In many kidney exchange programmes or kidney sharing schemes, the kidney travels from the donor centre to the recipient centre (as described in our recent study [38]), which obviously leads to increased CIT compared to LDKT not transplanted via a kidney sharing scheme. A strategy to keep the CIT to a minimum would be for the donor nephrectomy to be carried out in the same centre as the transplant, as practiced in the Netherlands and Canada, amongst others. In case of LDKT, this is relatively easy to facilitate, since live donors are healthy people who are perfectly capable of travelling (if willingly), with the downside that distances may be long, and it may be difficult for family to travel along and provide support. In addition, for kidney sharing scheme programmes in case of donor organs travelling to recipient centres, the CIT could be incorporated as a variable in the matching algorithm. If this is not an option and the kidney has to be transported over a long distance, one could investigate the use of donor organ machine perfusion. It is shown that hypothermic machine perfusion (HMP) reduces the risk of DGF and 1-year graft survival in deceased donor kidney transplantation [42,43], but evidence amongst LDKT is minimal and the only study performed did not detect any difference in DGF [44]. Future studies should explore the idea to deploy HMP during shipping of the kidneys with an expected prolonged CIT of more than four hours. Together, with existing literature on LDKT risk factors [11,29,38,44], it could be deemed cost-effective to support kidneys from high-risk donor–recipient pairs with HMP or deploy HMP during shipping of the kidney when a prolonged CIT of more than four hours is expected [45]. Studies included in the meta-analysis either did not mention machine perfusion [30,35] or excluded these patients [11,29], while only Nath et al. [31] discussed HMP and stated that it is attractive to establish the potential role of machine perfusion in LDKT because it reduces DGF and graft survival in DDKT, which we support. 

Another recent development is the increasing use of robotic surgery in living donor kidney transplantation, and we wish to discuss its consequences on CIT and LDKT. A recent meta-analysis [46] found excellent results when comparing robotic-assisted kidney transplantation to open kidney transplantation, but also found increased CIT in robot-assisted LDKT; they did not find a difference in DGF, not in acute rejection, renal function at six months and 1-year post kidney transplantation and not in graft or patient survival. The increased CIT probably has minimal clinical relevance since the mean difference was only 5.18 min (95% CI: 3.99–6.38, *p* < 0.001), which is positive because this implies that an increase in CIT is not a concern for robotic surgery. The prolonged CIT in robot-assisted kidney transplantation might be caused by a steep learning curve. Therefore, we hypothesize that these increased CITs are unlikely to persist in the future and will be diminished. Considering all factors, we agree with the authors that robotic surgery in LDKT is safe to further implement and that future advances in robotic surgery are not likely to increase the cold ischaemia time.

One of the most important strengths of our study is that it included a large patient cohort of 164,179 patients, and we feel that the large number of patients analysed in this review is a good representation since it is geographically widespread and includes studies from three continents. Furthermore, we performed a leave-one-out sensitivity analysis to estimate individual study effects which resulted in only one remarkable finding. We found a difference in the 1-year graft survival outcome variable comparing 0–2 h to 6–8 h of CIT when removing the study by Simkins et al. [35], which resulted in a significant *p* value in favour of 0–2 h of CIT. Acute rejection also became significant when the study by Nath et al. was removed from the meta-analysis. Some other forest plots show a high level of heterogeneity, for instance Figure 2, but the included studies in these forest plots are unambiguous regarding the outcome measure (the direction of the effect), and sensitivity analyses do not change the course of the outcome; the data are available in Appendix A.

### Limitations 

We wish to point out that there could be other factors contributing to the difference in studies which account for the heterogeneity amongst the studies. The first important factor is as to whether a kidney paired donation or a kidney exchange programme (KEP) exists in the country; these programmes match kidneys across a nation, which might lead to increased CIT; the CIT is significantly higher in countries where the kidneys travel rather than the donor in KEP programmes, for example the United Kingdom, the United States, and Australia. In the Netherlands and Canada, the donor travels to the recipient’s hospital (donor and recipient operations will be performed in the same hospital, associated with a shorter CIT). Not all included studies in this meta-analysis reported whether a prolonged CIT was caused by shipping of the kidney or due to another cause, such as intra-operative difficulties in the recipient. Most studies focus on LDKT in general and do not include sub-analyses for KEP versus non-KEP.

Another critical factor is that Krishnan et al. [30], Gill et al. [11], Nath et al. [31] and Simpkins et al. [35] excluded a CIT longer than eight hours, and Gill et al. excluded patients with a CIT longer than sixteen hours. Gill et al. excluded these patients because they were likely to be representing measure errors and because CIT longer than 16 h is uncommon in LDKT. Krishnan et al. excluded patients which experienced more than eight hours of CIT because of the likelihood of technical complications and uncertainty about the accuracy of the data. Nath et al. did not discuss the fact that only LDKT with CIT between zero and eight hours were included. Simpkins et al. excluded the LDKT with CIT longer than eight hours due to a broad distribution (median: 30 h, IQR: 20–40 h), the likelihood of technical complications and data entry errors. 

We know that pre-emptive transplantation results in benefits for the kidney recipient regarding patient survival and graft survival [8,47,48,49]. In every paper, except in Segev et al. [34], where pre-emptive transplantation was not specifically mentioned, a part of the study population (25–30%) underwent pre-emptive transplantation. In the study by Gill et al. [11], the 0–2 h CIT group contains only 4% pre-emptive transplantations and the other longer CIT groups all contain around 30% pre-emptive transplants (31%, 28% and 26%). These findings could affect the results of the forest plot analysis in that the graft survival, patient survival, and acute rejection are biased because a higher percentage of these groups represent pre-emptive transplantation. In all other studies, pre-emptive transplantation was evenly distributed over the different CIT groups, or only an overall percentage of pre-emptive transplants in the study was provided. 

Further, the risk of bias for our studies was scored as moderate, mainly because all studies categorized the CIT in groups of a certain number of hours (0–2, 2–4 h, etc.), which could, in theory, be changed later in the study to manipulate study results. Apart from this categorization, the studies scored a low risk of bias on almost all other aspects of the ROBIN-I risk of bias tool. All studies scored moderate on the bias of confounding aspect because there is a possibility that some variables are not considered in the multivariate regression analysis, an analysis performed by all studies, and residual confounding could arise. See Appendix A for a comprehensive overview of all variables accounted for. Especially, provider-related factors are not considered in the multiple regression analyses (experience with living donor KTx and hospital or surgeon volume for instance), meaning that residual confounding might be a problem. 

Some outcome variables scored a low certainty of evidence, estimated by the GRADE tool, and this is because these studies are retrospective cohort studies (which is a common finding for these types of studies when using this tool [50]). Unfortunately, there is no way to work around this because an RCT or prospective study would be unrealistic and even unethical. However, most studies scored a moderate level of evidence, and some scored high.

## 5. Conclusions

Our results, based on 164,179 patients, underline the need to keep CIT as short as possible in LDKT (ideally < 4 h), as a shorter CIT in LDKT is associated with a statistically significant lower incidence of DGF and higher graft survival compared to a prolonged CIT (>4 h). However, clinical impact seems limited, and therefore, in LDKT programmes in which the CIT might be prolonged, such as kidney exchange programmes, the benefits outweigh the risks [38]. To minimize these risks, it is worth considering including CIT in kidney allocation algorithms to reduce DGF, graft failure, and in general take precautions to protect high risk donor–recipient combinations.

## Figures and Tables

**Figure 1 jcm-11-01620-f001:**
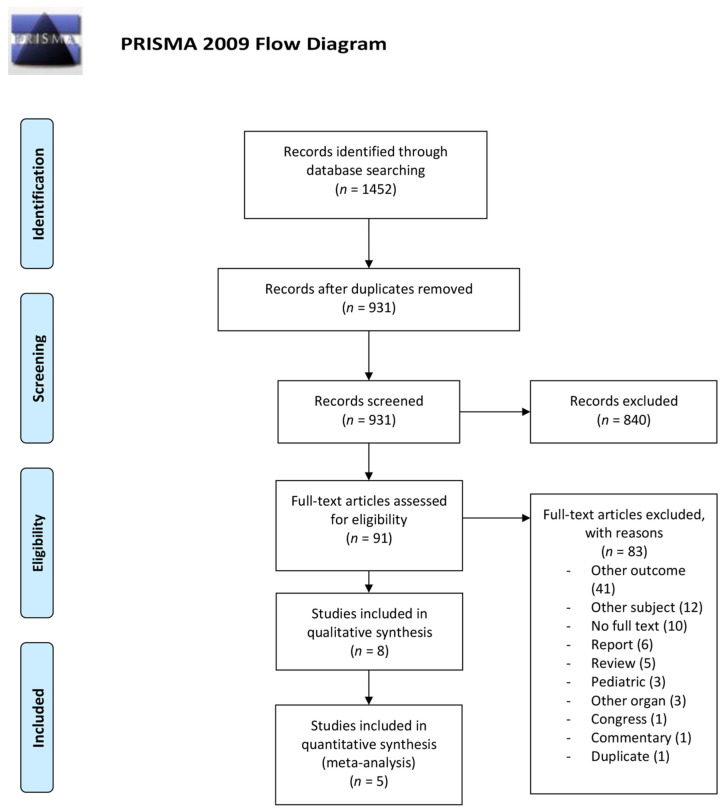
PRISMA (Preferred Reporting Items for Systematic Review’s and Meta-Analysis) flowchart of the systematic review search.

**Figure 2 jcm-11-01620-f002:**
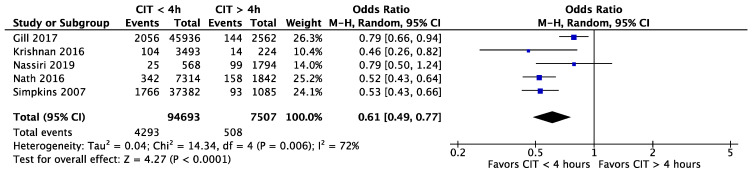
The incidence of DGF for CIT shorter and longer than 4 h. CIT: cold ischaemia time; DGF: delayed graft function.

**Figure 3 jcm-11-01620-f003:**
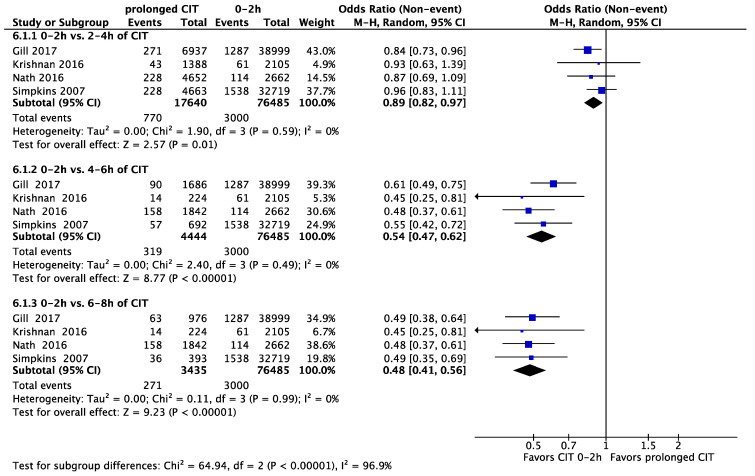
The incidence of DGF in 0–2 h (reference) of CIT versus 2–4, 4–6 and 6–8 h of CIT. CIT: cold ischaemia time; DGF: delayed graft function.

**Figure 4 jcm-11-01620-f004:**
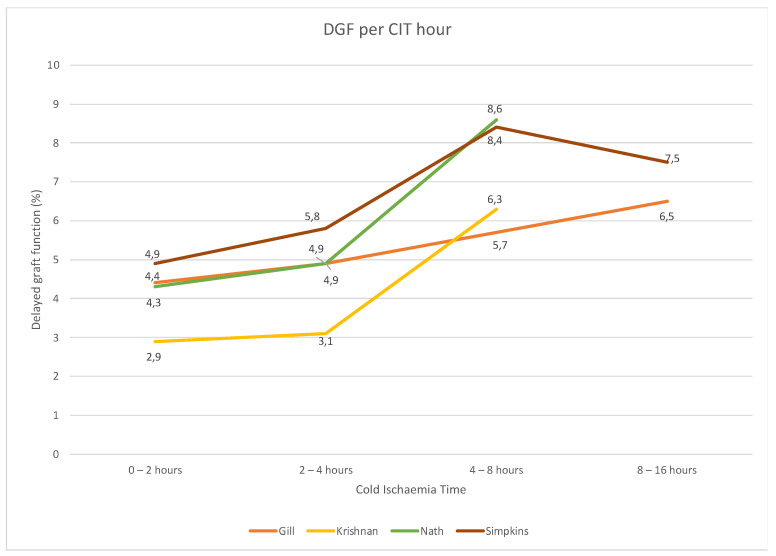
The incidence of DGF for 0–2, 2–4, 4–8 and 8+ hours of CIT. The studies included are Gill et al. [11], Krishnan et al. [30], Nath et al. [31] and Simpkins et al. [35]. CIT: cold ischaemia time; DGF: delayed graft function.

**Figure 5 jcm-11-01620-f005:**
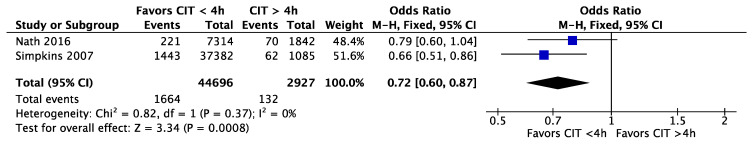
One-year graft survival for CIT shorter and longer than 4 h. CIT: cold ischaemia time.

**Figure 6 jcm-11-01620-f006:**
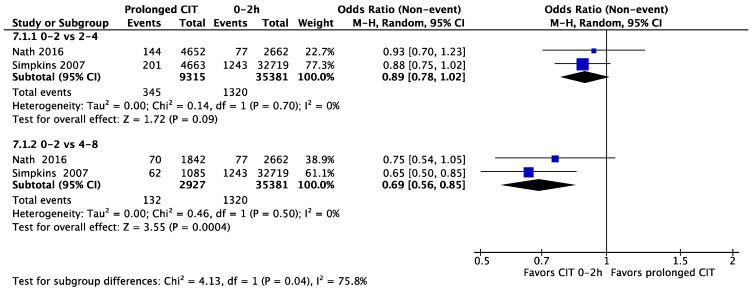
One-year graft survival; CIT of 0–2 h (reference) versus 2–4 and 4–8 h of CIT. CIT: cold ischaemia time.

**Figure 7 jcm-11-01620-f007:**
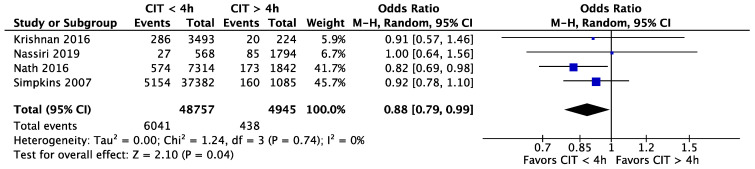
Five-year graft survival for CIT shorter and longer than 4 h. CIT: cold ischaemia time.

**Figure 8 jcm-11-01620-f008:**
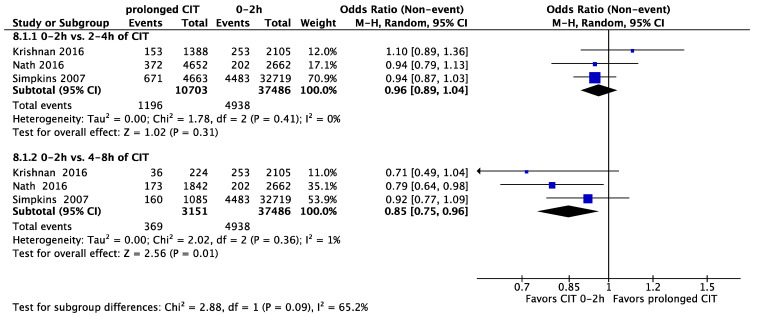
Five-year graft survival; CIT of 0–2 h (reference) versus 2–4 h and 4–8 h of CIT. CIT: cold ischaemia time.

**Figure 9 jcm-11-01620-f009:**
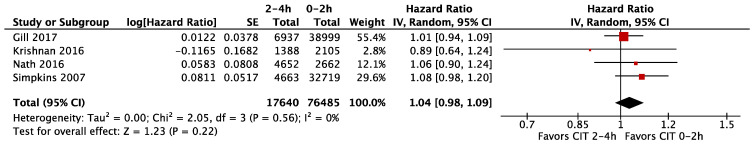
Five-year graft survival hazard ratio; CIT of 0–2 h versus 2–4 h. CIT: cold ischaemia time.

**Figure 10 jcm-11-01620-f010:**
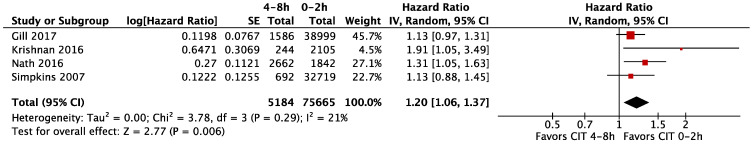
Five-year graft survival hazard ratio; CIT of 0–2 h versus 4–8 h. CIT: cold ischaemia time.

**Figure 11 jcm-11-01620-f011:**
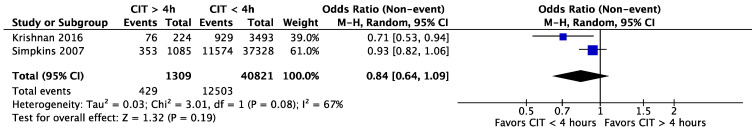
Ten-year graft survival for CIT shorter and longer than 4 h. CIT: cold ischaemia time.

**Figure 12 jcm-11-01620-f012:**
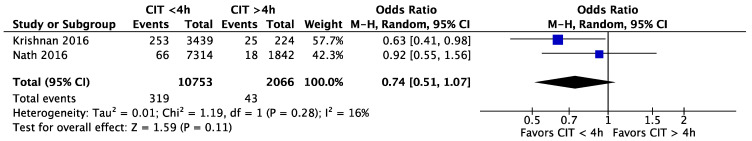
One-year patient survival for CIT shorter and longer than 4 h. CIT: cold ischaemia time.

**Figure 13 jcm-11-01620-f013:**
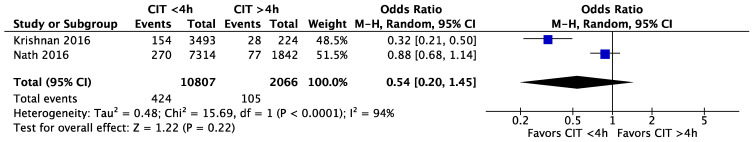
Five-year patient survival for CIT shorter and longer than 4 h. CIT: cold ischaemia time.

**Figure 14 jcm-11-01620-f014:**
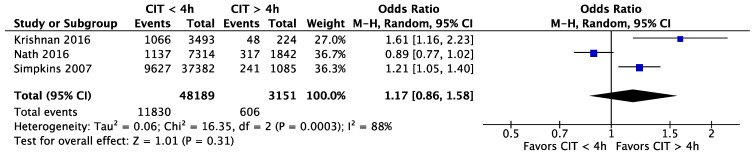
The incidence of acute rejection for CIT shorter and longer than 4 h. CIT: cold ischaemia time.

**Table 1 jcm-11-01620-t001:** Overview of the included studies in the systematic review.

	Year of Publication	Country	Study Design (Trial)	Intervention (*n*)	Control (*n*)	Total Patients	Follow-Up	Study Outcomes	ROBINS-I Tool (Risk of Bias)
Gill	2017	United States	retrospective	4–16 h (2562)	0–4 h (45,936)	48,498	m 4.53 year	DGF, GS	moderate
Krishnan	2016	Australia and New Zealand	retrospective	>4–8 h (224)	1–4 h (3493)	3717	m 6.6 year	DGF, AR, eGFR, GS, Mor, GF	moderate
Nassiri	2020	United States	retrospective	>16 h (141)	<16 h (2222)	2363	up to 7 years	DGF, DCGF	moderate
Nath	2016	United Kingdom	retrospective	>4–8 h (1842)	0–4 h (7314)	9156	-	GS (1, 3, 5 y), DGF, sCreat, PS, AR	moderate
Redfield	2016	United States	retrospective	DGF (2282)	no-DGF (61,760)	64,042	median 6^noDGF,^ 4^DGF^ days	DGF	moderate
Roodnat	2003	The Netherlands	prospective	≥12 h	<12 h	243	≥1 year	PS, GS, Mor	moderate
Segev	2011	United States	prospective	≥8 h (24)	<8 h (23)	56	-	UO	moderate
Simpkins	2007	United States	retrospective	4–8 h (1077)	0–4 h (37,390)	38,467	≥1 year	PS, GF, sCreat^1y^, DGF^1w^, AR^1y^, GS^10y^	moderate

DGF: delayed graft function; GS: graft survival; AR: acute rejection; eGFR: estimated glomular filtration rate; Mor: mortality; GF: graft function; sCreat: serum creatinine; PS: patient survival; UO: urine output; Creat: creatinine.

**Table 2 jcm-11-01620-t002:** Baseline characteristics of patients included in meta-analysis.

	Studies	0–4 h of CIT		4–8 h of CIT		*p* Value
		*n*		*n*		
Recipient age (mean (SD))	4 [11,30,31,35]	62,126	45.40 (2.82)	4737	45.27 (4.12)	0.92
Recipient sex (male) (%)	4 [11,30,31,35]	62,126	49.80	4737	53.99	0.92
Pre-emptive KTx (%)	3 [11,30,31]	56,744	12.28	3652	26.09	0.51
Donor age (mean (SD))	3 [11,30,31]	56,744	42.57 (4.22)	3652	41.90 (3.15)	0.94
Donor sex (male) (%)	3 [11,30,31]	56,744	39.87	3652	41.90	0.99
HLA-mismatch (1–3) (%)	2 [11,31]	53,251	46.31	3428	58.13	0.99
HLA-mismatch (4–6) (%)	2 [11,31]	53,251	41.81	3428	39.33	0.82
Diabetes ESRD (%)	4 [11,30,31,35]	62,126	18.30	4737	15.15	1.00
Peak PRA = 0 (%)	2 [11,31]	53,251	59.05	3428	47.03	0.06
Peak PRA > 80 (%)	2 [11,31]	53,251	5.93	3428	16.70	0.36

CIT: cold ischaemia time, ESRD: end-stage renal disease, HLA: human leucocyte antigen, PRA: panel reactive antibody, KTx: kidney transplantation.

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
