# Peer review of "The Impact of Cold Ischaemia Time on Outcomes of Living Donor Kidney Transplantation: A Systematic Review and Meta-Analysis"

_jcm, 2022, doi:10.3390/jcm11061620_

Round 1

Reviewer 1 Report

In this study the authors report a systematic review and met-analysis aiming to assess the impact of cold ischemia time (CIT) on key clinical outcomes after living-donor kidney transplantation.

Despite not entirely novel, the topic of the study is timely and interesting for readers of the Journal of Clinical Medicine. Moreover, the review may provide several insights for future research in this field.

The following aspects should be addressed/discussed by the authors to improve the quality of the manuscript and to clarify a few aspects of concern related to the review design.

  1. The choice to perform a “meta-analysis” is questionable given the lack of RCTs and the high proportion of retrospective studies (with high risk of bias and confounding). As such, the authors may consider re-naming the meta-analysis as a “quantitative synthesis” focusing on a few outcomes for whom the risk of bias could be minimal.
  2. Please discuss the potential impact of residual confounding on the results of quantitative analysis. In fact, the impact of several provider-related factors (hospital/surgeon volume and experienced with LD kidney transplantation, etc.) on DGF and graft survival could not be analyzed. The same concept applies to several donor- and recipient-related factors. How could the authors be confident that the analysis on the impact of CIT on graft/patient survival was not influenced by other donor-, recipient- and provider-related factors?
  3. Please expand the discussion on the potential technical and logistical strategies to minimize CIT in the setting of LD kidney transplantation.
  4. Please provide a Table with the complete RoB assessment for each individual study included in the qualitative and quantitative analysis.
  5. Please discuss the potential role of robotic surgery for living-donor kidney transplantation and its potential impact on the outcomes of interest for the review.

Author Response

Please see attachment. References are for all reviewers, which is why not all references may be present in the letter.

Reviewer 2 Report

Thank you for the opportunity to review the manuscript by van de Laar et al. titled “The Impact of Cold Ischaemia Time on Outcomes of Living Donor Kidney Transplantation: A Systematic Review and Meta-Analysis”.

In their meta-analysis, the authors aimed to investigate the effect of a cold ischemia time (CIT) on the outcomes of living donor kidney transplantation (LDKT). In their unrestricted literature search until July 2021, the authors identified articles that compared different CIT in LDKT and described delayed graft function (DGF), graft- and patient survival and acute rejection. Out of 1,452 articles, the authors included 8 eligible studies of 164,179 patients. They showed significantly lower incidence of DGF and higher 1-year graft survival, and 5-year graft survival if the CIT was below 4 h. The authors concluded that a CIT < 4 h in LDKT favors better outcomes after LDKT and should be kept below 4 h where possible. The authors also suggested that CIT should be considered in the matching algorithm of kidney paired exchange programs.

The authors are to be commended for the idea and for the well-structured manuscript, I have, however, some remarks.

I am not quite convinced that the poor long-term graft survival is just because of longer CIT. A recent analysis of the impact of CIT on outcome following deceased donor liver transplantation revealed that the duration of cold storage becomes irrelevant after the first year following transplantation (Front Immunol. 2020 May 12;11:892. doi: 10.3389/fimmu.2020.00892. eCollection 2020.). Similarly, every additional hour of CIT increases the risk of 1-year liver graft loss, but the negative effect depends highly on the indication for liver transplantation. Being a more immunogenic organ than the liver, the outcomes of kidney transplantation depend highly on the immune systems of donor and recipient, but also on other factors, such as immunosuppression therapy, PRA etc.. The authors should elaborate and explain the poor 5-year graft survival extensively. However, this may be difficult because of the low number of studies included in the meta-analysis, which is why the conclusion should not be that optimistic.

I presume Population, Intervention, Comparison, Outcome, Time and Study design (PICOTS) strategy was used to select studies. The authors should define and elaborate the eligibility and inclusion criteria in the Methods section more extensively.

It would be interesting to see funnel plots to assess publication (reporting) bias.

Minor comments

The manuscript needs to be checked for typos.

The sentence on page 7, line 161 is incomplete.

The authors are encouraged to avoid statements such as “to this date, no systematic review or meta-analysis had been” or “this is the first study” etc.

The authors should reduce the use of acronyms to a minimum and provide a legend of the used ones. This would render the manuscript easier to read.

Reviewer 3 Report

Very interesting, rigorous and well written paper

I have nothing to say but…congratulations

Round 2

Reviewer 1 Report

All questions and remarks have been sufficiently answered. The corrections that have been made improved the quality of the article. I have no further comments.

Reviewer 2 Report

The authors have substantially improved their manuscript after revision. I have no further remarks.

Best regards